# Weighted Iterative CD-Spline for Mitigating Occlusion Effects on Building Boundary Regularization Using Airborne LiDAR Data

**DOI:** 10.3390/s22176440

**Published:** 2022-08-26

**Authors:** Renato César dos Santos, Ayman F. Habib, Mauricio Galo

**Affiliations:** 1Department of Cartography, São Paulo State University (UNESP), Presidente Prudente, São Paulo 19060-900, Brazil; 2Lyles School of Civil Engineering, Purdue University, West Lafayette, IN 47907-2050, USA

**Keywords:** boundary modeling, LiDAR, vegetation occlusion, remote sensing, urban application

## Abstract

Building occlusions usually decreases the accuracy of boundary regularization. Thus, it is essential that modeling methods address this problem, aiming to minimize its effects. In this context, we propose a weighted iterative changeable degree spline (WICDS) approach. The idea is to use a weight function for initial building boundary points, assigning a lower weight to the points in the occlusion region. As a contribution, the proposed method allows the minimization of errors caused by the occlusions, resulting in a more accurate contour modeling. The conducted experiments are performed using both simulated and real data. In general, the results indicate the potential of the WICDS approach to model a building boundary with occlusions, including curved boundary segments. In terms of *F_score_* and *PoLiS*, the proposed approach presents values around 99% and 0.19 m, respectively. Compared with the previous iterative changeable degree spline (ICDS), the WICDS resulted in an improvement of around 6.5% for completeness, 4% for *F_score_*, and 0.24 m for the *PoLiS* metric.

## 1. Introduction

The extraction of building boundaries is an important task in urban applications such as 3D city modeling, disaster management, database updating, and urban planning. In general, mapping companies and municipalities derive building boundaries from topographic surveys, manual image vectorization, or restitution processes. However, they are time consuming, particularly in large urban areas and constantly changing regions. Considering these aspects, the scientific community has turned its efforts to develop automatic or semi-automatic techniques for deriving building boundaries from remotely sensed data.

In this context, airborne LiDAR data have been widely used since they have some advantages when compared with conventional photogrammetry. The main advantage is related to the direct acquisition of dense 3D point clouds, which forgoes the need for an image matching stage. Additionally, LiDAR data are not affected by scene conditions such as shadows and illumination. Previous studies have looked at combined airborne LiDAR with aerial or satellite images. Despite the improvement in extracted contours, the integration is still faced with some challenges due to the varying nature of these datasets (e.g., regular versus irregular data structure and ensuring the alignment of both datasets to a common reference frame) [1,2].

According to dos Santos et al. [1], building boundaries derived from LiDAR data have an aliasing shape (zigzag); thus, a regularization/modeling process is usually applied to obtain a contour closer to the real boundary. The vast majority of regularization methods are based on building boundaries that are made up of straight-line segments [3,4,5,6,7,8,9,10,11,12]. In this sense, developed methods estimate line parameters that best fit and represent each segment. Additionally, parallelism and/or perpendicularity constraints are applied. In [13], the building regularization is performed through a Recursive Minimum Bounding Rectangle (RMBR) algorithm, which determines the rectangle or combination of rectangles that best fits the boundary points. In contrast, there are approaches that consider more complex building boundaries, which have non-right-angled corners [1,4,14] and curved segments [1,14]. Dos Santos et al. [1] developed an iterative changeable degree spline (ICDS) regularization method in 3D space. In their approach, the polynomial function that best models each segment is estimated automatically through a statistical analysis. Conducted experiments indicated that the method is robust for modeling contours with complex curved segments. Despite the promising results, the modeling is strongly influenced by occlusions.

Figure 1 shows a building partially covered by a tree, as well as a corresponding point cloud and modeled boundary obtained by applying the ICDS approach (blue contour). As can be observed in Figure 1c, part of the roof building is missing due to tree canopy occlusion, leading to an incorrect contour modeling (Figure 1d).

According to Feng et al. [2], in an urban environment, many buildings might be partially covered by adjacent trees so that the laser cannot pass through when the leaves are relatively dense, especially during the spring and summer. To overcome this problem, the authors proposed an improved minimum bounding rectangle (IMBR) algorithm to extract and model building boundaries with partial occlusion from airborne LiDAR data. The IMBR algorithm is executed in two-dimensional space, looking to model buildings composed of straight segments and with right-angled corners.

To overcome the aforementioned limitation, we propose a weighted iterative changeable degree spline method (WICDS). The idea consists of including the occlusion information in boundary modeling and assigning a lower weight to contour points located in the occlusion region.

The main contribution of this work is to propose a novel approach, which allows automatic modeling of boundaries with partial occlusions even for curved contours. Additionally, it is executed in 3D space, making it possible to model the 3D spatial shape of the contour. To perform qualitative and quantitative evaluation, simulated occlusions together with real airborne LiDAR data are used. In addition, we also evaluate the influence of occlusion size and weight magnitude on contour modeling.

## 2. Proposed Method

In Figure 2, we show a simplified flowchart of the proposed approach. The blue dashed rectangle highlights the steps that differ from the ICDS approach proposed in [1]. The building roof points are obtained using the same strategy of the previous approach ICDS and since the focus of this paper is on contour modeling, the occlusion regions were identified manually by visual inspection. The occlusion information is incorporated in three steps: critical point determination, changeable degree (CD) spline modeling, and determination of residuals. Similar to the ICDS approach, we use the residuals for automatic selection of the polynomial function.

### 2.1. Critical Point Determination

Considering the boundary points in 3D space, critical (key) points are obtained using the well-known Douglas–Peucker algorithm, followed by angle-based generalization. Similar to [1,14], Douglas–Peucker algorithm is executed in 3D space using a distance threshold ($T_{\mathrm{dist}}$), whereas the elimination of redundant critical points is executed using an angle threshold ($T_{\mathrm{ang}}$). In the angle-based generalization step, we calculate the angles between two adjacent lines formed by connecting adjacent critical points. If the angle (*θ_i_*) is smaller than an angle threshold ($T_{\mathrm{ang}}$), the point is regarded as a redundant point and discarded. In Figure 3c, we illustrate the geometric representation of the angle *θ*.

The selection of the critical points depends on the thresholds, especially the $T_{\mathrm{ang}}$ as discussed in the previous work [1]. The value of $T_{\mathrm{dist}}$ can be easily determined based on point cloud spacing. In contrast, the $T_{\mathrm{ang}}$ may vary according to the complexity of the building, requiring a priori knowledge of the buildings contained in the area of interest.

Occlusions will cause loss of information on part of the roof, consequently affecting the critical point determination (Figure 3b). In this sense, we use the occlusion information to identify and eliminate critical points extracted in impacted regions.

In Figure 3, we show an example of critical point determination for a partially occluded building roof. The boundary points are extracted using the adaptive alpha-shape algorithm [15]. The points in the occlusion region, highlighted by the cyan dashed rectangle, are represented by orange square points. In Figure 3b,d,e, we show the results derived from the Douglas–Peucker algorithm, angle-based generalization, and occlusion-based refinement, respectively.

### 2.2. CD-Spline Modeling and Weight Function

Considering the boundary points as input data, the CD-spline is applied to model a parametric curve. According to [16], the mathematical model represented by C(t) (Equation (1)) is a piecewise polynomial function used to model a curve through the boundary points, whereas D is the largest degree considered in the modeling process ($D=\max\left\{d_{i}\right\}$). This equation is composed of two terms. The first term represents the CD-splines basis functions (*N_i,D_*). The elements of *N_i,D_* are computed from the degree of the polynomial (*d_i_*), critical points (knot points), and continuity type between connected adjacent segments. The second term (*P_i_*) is formed by control points, which controls the shape of the curve and is determined through an estimation process, using the least-squares method, for example.
(1)$$C\left(t\right)=\overset{n}{\underset{i=1}{\sum }}N_{i,D}\left(t\right)P_{i}$$

Assuming *m +* 1 boundary points in 3D space, i.e., ($Q_{0}{,\;Q}_{1},\;......{\;,\;Q}_{m}$), these points can be represented by a vector $Q_{r}$ (Equation (2)), with $Q_{j}=[x_{j}{\;y}_{j}{\;z}_{j}]$.
(2)$$Q_{r}{=\{Q}_{0}{,\;Q}_{1},\;......{\;,\;Q}_{m}\}$$

The ICDS approach uses the chord length formulation to parameterize the coordinates of the piecewise polynomial function *C(t)* (Equations (3) and (4)):(3)$$t_{0}{=0\;t}_{j}{=t}_{j-1}+\frac{\left\|Q_{j}-{\;Q}_{j-1}\right\|}{L}{\;t}_{m}=1$$
where
(4)$$L=\overset{m}{\underset{j=1}{\sum }}\left\|Q_{j}-{\;Q}_{j-1}\right\|$$

Considering the occurrence of occlusions, we propose the following parameterization (Equations (5)–(8)):(5)$$\{\begin{array}{c} t_{0}{=0,\;t}_{m}=1\\ t_{j}{=t}_{j-1}+\frac{\left\|Q_{j}-Q_{j-1}\right\|}{L}\;\to \;\forall \;\mathrm{point}\;j\;\mathrm{outside}\;\mathrm{occlusion}\;\mathrm{region}\;\\ t_{j}{=h}_{j}\left(\frac{t_{{\mathrm{final}}_{\mathrm{occl}}}-t_{{\mathrm{initial}}_{\mathrm{occl}}}}{h_{\mathrm{final}}-h_{\mathrm{initial}}}\right)+\left(\frac{h_{\mathrm{final}}{\;t}_{{\mathrm{initial}}_{\mathrm{occl}}\;}-h_{\mathrm{initial}}{\;t}_{{\mathrm{final}}_{\mathrm{occl}}}}{h_{\mathrm{final}}-h_{\mathrm{initial}}}\right)\;\to \;\forall \;\mathrm{point}\;j\;\mathrm{in}\;a\;\mathrm{given}\;\mathrm{occlusion}\;\mathrm{region} \end{array}$$
where
(6)$$h_{\mathrm{initial}}=\|Q_{{\mathrm{initial}}_{\mathrm{occl}}}-{\;Q}_{{\mathrm{initial}}_{\mathrm{occl}}-1}\|$$
(7)$$h_{\mathrm{final}}=\overset{{\mathrm{final}}_{\mathrm{occl}}}{\underset{{j=\mathrm{initial}}_{\mathrm{occl}}}{\sum }}\left\|Q_{j}-{\;Q}_{j-1}\right\|$$
(8)$$h_{j}{=h}_{j-1}+\|Q_{j}-Q_{j-1}\|$$

In Figure 4, we show the representation of the points $t_{{\mathrm{initial}}_{\mathrm{occl}}}$ and $t_{{\mathrm{final}}_{\mathrm{occl}}}$ for a contour with occlusion. In this example, the contour points are organized in a clockwise order. It is important to emphasize that a building with multiple occlusions will also have multiples $t_{{\mathrm{initial}}_{\mathrm{occl}}}$ and $t_{{\mathrm{final}}_{\mathrm{occl}}}$.

According to [17], the control points are estimated from a least-squares adjustment process. In this context, the error function in Equation (9) is considered [17]. To determine the control points, the ICDS approach uses the formulation shown in Equation (10).
(9)$$\phi =\overset{m}{\underset{j=0}{\sum }}\left|Q_{j\;}-{\;C(t}_{j})\right|$$
(10)$${P=(A}^{T}{A)}^{-1}{\;(A}^{T}Q_{r})$$
where *Q_j_* corresponds to a given contour point in 3D space (*Q_j_ = [x_j_ y_j_ z_j_]*), *Q_r_* = {*Q*_0_, *Q*_1_, …, *Q_m_*}, and *A* is the Jacobian matrix defined by the basis function elements.

ICDS approach assumes that all contour points have equal weight in the CD-spline modeling (*w*_0_
*= w*_1_ = *w*_2_
*= … = w_m_*). Additionally, in this approach, the variance related to each point ($\sigma_{j}^{2}$) is equal to an a priori variance factor ($\sigma_{0}^{2}$). From these two assumptions, the weight matrix (*W*) can be represented by an identity matrix (*I*) (*W = I*). However, this strategy can lead to incorrect modeling when building roofs are partially occluded (Figure 1d).

Usually, boundary points extracted in occluded regions negatively influence the contour modeling, as exemplified in Figure 1d. To minimize this problem, we propose the inclusion of a weight function, which assigns a lower weight to boundary points in the occlusion region. In Equation (11), we show the formulation of the weight function for points located outside and inside occluded regions:(11)$$\{\begin{array}{c} w_{j}=\frac{\sigma_{0}^{2}}{\sigma_{\mathrm{LiDAR}}^{2}}\;\to \;\forall \;\mathrm{point}\;j\;\mathrm{outside}\;\mathrm{occlusion}\;\mathrm{region}\;\\ w_{j}=\frac{\sigma_{0}^{2}}{\sigma_{\mathrm{Oclussion}}^{2}}\;\to \;\forall \;\mathrm{point}\;j\;\mathrm{in}\;\mathrm{occlusion}\;\mathrm{region} \end{array}$$
where $\sigma_{\mathrm{LiDAR}}^{2}={\;\sigma }_{\mathrm{xy}}^{2}+{\;\sigma }_{z}^{2}$, and $\sigma_{\mathrm{Oclussion}}^{2}$ = *b* $\sigma_{\mathrm{LiDAR}}^{2}$.

The terms $\sigma_{\mathrm{xy}}^{2}$ and correspond to the approximate variance of the LiDAR data in planimetric and vertical directions, respectively, and *b* is a multiplying factor, which can vary from 1 to +∞. Assuming $\sigma_{0}^{2}$ = $\sigma_{\mathrm{LiDAR}}^{2}$, the weight function is simplified for the following formulation (Equation (12)):(12)$$\{\begin{array}{c} w_{j}=1\;\to \;\forall \;\mathrm{point}\;j\;\mathrm{outside}\;\mathrm{occlusion}\;\mathrm{region}\;\\ w_{j}=\frac{1}{b}\;\to \;\forall \;\mathrm{point}\;j\;\mathrm{in}\;\mathrm{occlusion}\;\mathrm{region} \end{array}$$

With the inclusion of the weight function, Equation (13) is used to estimate the control points.
(13)$${P=(A}^{T}{\mathrm{WA})}^{-1}{\;(A}^{T}{\mathrm{WQ}}_{r})$$

### 2.3. Residual Determination for Boundary Points in Occluded Regions

In the ICDS approach, the polynomial function that best models each segment is automatically selected by an incremental and iterative process, where the stopping criterion is based on the statistical *F*-test. In the first iteration, all segments are modeled by a first-degree polynomial. After this modeling, the sum of residuals for each segment is determined. In the next iteration, the degree of the function (*d_i_*) corresponding to the segment with the biggest sum of residuals is increased by one (*d_i_ +* 1). In the following iterations, the boundary is modeled considering the new polynomial degree. The process is repeated until there is no significant difference between the modeled boundaries between two successive iterations.

The magnitude of the residual (*r_j_*) for each point and the sum of residuals for each segment (*Sum_r_i_*) are computed from Equations (14) and (15), respectively:(14)$$r_{j}=\left|{C(t}_{j})\;-{\;Q}_{j}\right|$$
(15)$${\mathrm{Sum}\_r}_{i}=\sum r_{j}^{i}$$

The statistical *F*-test (Equation (16)) is conducted using the estimated statistic ($F_{c}$*)* (Equation (17)) [18,19], which is obtained considering the standard deviations of residuals in iterations *k* and *k −* 1. In this test, *F_c_* is compared to the critical values of the *F* distribution ($F_{\frac{\alpha }{2},\;n-1,\;n-1}$, $F_{1-\frac{\alpha }{2},\;n-1,\;n-1}$) based on the number of boundary points (*n*) and level of significance (*α*):(16)$$\{\begin{array}{c} F_{\frac{\alpha }{2},\;n-1,\;n-1}{\;<\;F}_{c\;}{\;<\;F}_{1-\frac{\alpha }{2},\;n-1,\;n-1}\;\to \;\mathrm{process}\;\mathrm{is}\;\mathrm{finished}\\ \mathrm{otherwise}\;\to \;\mathrm{go}\;\mathrm{to}\;\mathrm{next}\;\mathrm{iteration} \end{array}$$
where
(17)$$F_{c}={\left(\frac{s_{r}^{k}}{s_{r}^{k-1}}\right)}^{2}$$

Since ICDS took into consideration the residuals of all contour points, an overfitting problem may occur in occluded regions. To overcome this problem, we include an exception in the residual estimation (Equation (18)). In this case, we adopt *r_j_* = 0 for boundary points located in the occlusion regions:(18)$$\{\begin{array}{c} r_{j}=0\;\to \;\mathrm{if}\;\mathrm{point}\;j\;\mathrm{is}\;\mathrm{located}\;\mathrm{in}\;\mathrm{occlusion}\;\mathrm{region}\\ r_{j}=\left|{C(t}_{j})\;-{\;Q}_{j}\right|\to \;\mathrm{otherwise} \end{array}$$

## 3. Experiment Design and Quality Assessment

In the experiments, we consider simulated and real data. In the first case, we simulated occlusions of different sizes for two buildings: a rectangular building and another composed of straight-line and curved segments. Occlusions were manually generated using Cloud Compare software (https://www.cloudcompare.org, accessed on 15 January 2022). In the second case, we selected buildings with partial occlusions from the Presidente Prudente/Brazil dataset [20]. The 3D point cloud used in the experiments has a point density of around 12 points/m^2^. It was acquired from an average flying height of 550 m above ground using a REIGL LMS-Q680i scanning system [20]. To perform the quantitative analysis, the modeled contour is compared with the reference boundary. The reference boundaries are derived manually from airborne LiDAR using Cloud Compare software.

To carry out a quantitative analysis, we use the following metrics: relative error in area, completeness, correctness, *F_score_* [21,22], and *PoLiS* [23]. The relative error in area (*E_R_*) is estimated using the values of area of reference (*A_R_*) and the area estimated from the extracted contour (*A_E_*) (Equation (19)).
E_R_ = (A_E_ − A_R_)/A_R_(19)

Assuming an extracted polygon (A) and the reference polygon (B), completeness (*Comp.*), correctness (*Corr.*), and *F_score_* can be obtained from Equations (20)–(22) [21,22]:(20)Comp=ar(TP)/(ar(TP)+ar(FN))
(21)Corr=ar(TP)/(ar(TP)+ar(FP))
(22)$$F_{score}=2\;\mathrm{ar}(\mathrm{TP})/(2\mathrm{ar}(\mathrm{TP})+\mathrm{ar}(\mathrm{FP})+\mathrm{ar}(\mathrm{FN}))$$
where *ar*(.) is the measured area; *ar(TP) = A∩B; ar(FN) = ar(B) − A∩B; and ar(FP) = ar(A)* − *A∩B*; true positive (*TP*); false positive (*FP*); and false negative (*FN*). The completeness, correctness, and *F_score_* are estimated in 2D space and range from 0 to 1. Values approaching 1 indicate that the modeled boundary has a high overlap with the reference contour.

The *PoLiS* metric p(A, B) between two polygons A and B is defined by Equation (23) [23]:(23)$$p\left(A,\;B\right)=\frac{1}{2q}\;\underset{a_{j}\in A}{\sum }\underset{b\in \partial B}{\min}\left\|a_{j}-b\right\|+\frac{1}{2r}\;\underset{b_{k}\in B}{\sum }\underset{a\in \partial A}{\min}\left\|b_{k}-a\right\|$$
where q and r correspond to the number of vertices of polygons A and B, respectively; ∂A and ∂B denote the boundary of polygons A and B, respectively; and ‖a−b‖ is the Euclidean distance between points a and b. The *PoLiS* is computed in 3D space and ranges from 0 to +∞, approaching zero if the modeled boundary is closely similar to the reference.

## 4. Results

### 4.1. Simulated Data

In Figure 5 and Figure 6, we show the buildings with simulated occlusions (first row), as well as results derived from the ICDS (second row) and WICDS (third row). The modeled contour is represented by blue lines. In addition, the occlusion size is indicated in the figures. These results of WICDS are generated using *w* = 1/300 (*b* = 300 in Equation (12)).

In Figure 7, we show the quality metrics *F_score_* and *PoLiS* for the simulated buildings using both modeling methods (ICDS and WICDS).

In order to evaluate the influence of weight, we show the plots in Figure 8. Several weight values, i.e., different values of *b*, are adopted for modeling buildings with partial occlusions. In total, fourteen values are considered (*w =* 1; *w =* 1/5; *w =* 1/10; *w =* 1/50; *w =* 1/100; *w =* 1/200; …; *w =* 1/1000). Figure 8a,b show the *F_score_* values corresponding to each weight for rectangular and curved buildings, respectively, whereas Figure 8c,d show the *PoLiS* values. In addition, we show the visual result of the modeling for buildings B1_oc2 and B2_oc2 using different weight values (Figure 9).

### 4.2. Real Data

In order to evaluate the proposed method for a real environment, we selected a set of buildings from the Presidente Prudente/Brazil dataset (Figure 10, Figure 11, Figure 12, Figure 13 and Figure 14). In this case, all buildings have occlusions caused by nearby trees. Care was taken in the selection, aiming to choose buildings and occlusions with different complexities. Similar to the simulated dataset, the WICDS was performed using *w* =1/300.

In Figure 10, we show the results using the ICDS and WICDS methods for buildings B3–B8. Additionally, we show aerial image patches corresponding to each building and its surroundings, as well as the points sampled on the building roof. In order to show the influence of weight on real data, we show the modeling results for different weight values for buildings B6 and B7 (Figure 11).

In Figure 12, we show building B8 considering its projection in 2D space. This building has a pitched roof (Figure 10) and the occlusion covers part of the ridge up to the rightmost corner. The orange rectangle highlighted the segment with occlusion, as well as the results derived from both modeling methods (Figure 12).

In Figure 13, we show the 2D and 3D representations for buildings B9–B11. In this case, building B9 is partially occluded by two trees, whereas building B10 is partially occluded by a tree and three circular antennas (right side). Building B11 has a small occlusion caused by a large antenna. In Figure 13a, the cyan rectangles highlight the occlusions caused by these antennas.

Figure 14 shows two cases where the occlusion occurs at the building corner. In buildings B12 and B13, both occlusions are caused by nearby trees. In this case, B12 is composed of curved segments, whereas B12 is composed only of straight-line segments. The orange rectangles highlight the modeling at the occlusion region.

In Table 1, we show the reference area for selected buildings, as well as the area and relative error corresponding to modeling approaches. In Table 2, we show the completeness, correctness, *F_score_*, and *PoLiS* metrics for buildings B3–B11 using ICDS and WICDS methods. In the last column in Table 2, we show the metric *PoLiS* corresponding to WICDS and the percentage of improvement or deterioration with respect to the ICDS. In Figure 15, we show graphics of the quality metrics *F_score_* and *PoLiS* for buildings B3–B11 considering both approaches.

## 5. Discussion

Through a visual analysis of modeled boundaries for the simulated dataset (Figure 5 and Figure 6), we can observe that contours are not correctly modeled by the ICDS approach in occlusion regions. In building B1, the ICDS adjusted curves of different degrees for the occlusion regions (B1_oc1, B1_oc2, and B1_oc3), incorrectly modeling the irregularity caused by occlusion. In building B2, the ICDS properly models the curved segment when a small occlusion is present (B2_oc1). However, this is not the case for larger occlusions (B2_oc2 and B2_oc3). In contrast, the WICDS correctly models all segments, even when dealing with occlusions with different magnitudes. The exception occurred in building B3_oc3, which has a large occlusion covering (around 70%) of a curved segment. In this case, it is modeled as a straight-line segment instead of a curve. In general, this visual analysis indicates the potential of the proposed method in modeling contours with different magnitudes of occlusion, including those buildings composed of curved segments.

Analyzing the *F_score_* and *PoLiS* plots for buildings B1 and B2 (Figure 7), we can observe that modeling using ICDS tends to get worse with increasing occlusion sizes. For the WICDS, the quality metrics remained stable even when experiencing occlusions of different magnitudes, having *F_score_* and *PoLiS* values very close to buildings without occlusions. Additionally, it is possible to note that only the *F_score_* of curved buildings with large occlusions (B2_oc3) has a large difference, since the curved segment is modeled by a straight line. These results reinforce the potential of WICDS and its superiority over the ICDS.

In Figure 8, we show the *F_score_* and *PoLiS* plots for buildings with partial occlusions as a function of the weight values, i.e., by changing the multiplying factor *b* (Section 2.2). It is possible to note that weight value may influence the boundary modeling. However, the results tend to stabilize when a weight value less than 1/100 is considered for points located at the occlusion region, as can be observed in plots. This behavior is also observed in Figure 9 and Figure 11, where the visual representation of the modeled contour using different weights is shown. This information can guide the user in defining the weight.

Considering buildings B3–B8 from the Presidente Prudente/Brazil dataset (Figure 10), it is possible to note that most of the occluded segments are not correctly modeled by ICDS, especially those with large occlusions, for example, buildings B4–B6. The ICDS has a problem in the modeling of curved buildings with occlusions (building B7). It can observe that the two curved segments are modeled by a straight line. In contrast, the WICDS method correctly modeled the contours in most cases, including buildings B4–B6 with a high occlusion level and curved building B7. The proposed method has a problem in modeling the corner of the pitched roof in building B8, since the occlusion occurred in this region.

Although the proposed approach does not model the corner of the pitched roof in B8, the contour in 2D space has a consistent representation, as can be observed in Figure 12. In contrast, the ICDS method is able to extract the corner of the pitched roof: however, the final contour has an irregular shape in the occlusion region (Figure 12).

In buildings B9–B11, it is possible to observe multiple occlusions, which are caused by trees and circular antennas (Figure 13). Even with this complex scene, all contours are correctly modeled by the WICDS method. In contrast, the ICDS approach is not able to properly model the segments with tree canopy occlusions in buildings B9 and B10, and the segment with an antenna occlusion in building B10.

Performing a visual analysis in Figure 14, it is possible to observe that both methods present problems to model contours when the occlusion occurs close to the building corners. In the case of ICDS, the method tries to adjust small straight-line segments in the occlusion region. In the WICDS approach, the occlusion is modeled by a curved segment. In fact, this is one critical situation and these results indicate a limitation of both methods to model contours with occlusion in corner regions.

In Table 1, it can be seen that the relative error in the area tends to be negative, i.e., the extracted area is smaller than the reference area. This characteristic is related to the subsampling of LiDAR data. Comparing the two approaches, it is possible to observe that WICDS has a smaller magnitude of relative error for the majority of buildings.

In terms of completeness (Table 2), the WICDS presents an average value of 97.8%, against 91.3% of the ICDS. In this case, the WICDS presents an improvement of around 6.5% in completeness. It is important to highlight that the completeness of the ICDS approach is influenced by the size of the occlusion, i.e., higher occlusions lead to lower completeness values, as can be observed in Table 2. Considering the correctness values, both methods present values of 99.2%. Analyzing the *F_score_* metric, the WICDS has an average value of around 98.5%, against 94.8% of the ICDS. This increase in *F_score_* reflects the improvement in completeness, since *F_score_* is a harmonic mean between completeness and correctness. In general, these results indicate the improvement of the proposed method in terms of completeness for the modeling of boundaries with occlusions.

In terms of the *PoLiS* metric, the ICDS has a value of around 0.43 m, against 0.19 m of the WICDS. It represents an increase of around 0.24 m (improvement of ≈56%). Building B7 has the greatest magnitude of improvement going from 1.37 to 0.10 m (≈92%). These results indicate the robustness of the proposed method when working with partially occluded buildings.

Analyzing the *F_score_* and *PoLiS* plots for buildings B3–B11 (Figure 15), we observe that WICDS presents similar or better values than ICDS. The exception is the *PoLiS* value for building B8, since the proposed approach is not able to extract the corner point of a pitched roof.

In summary, the qualitative and quantitative analysis indicates that the proposed strategy has the potential to model building boundaries with occlusions, including boundaries formed by curved segments and occlusions of different sizes.

## 6. Conclusions

This paper proposes a method for modeling building boundaries with partial occlusions from airborne LiDAR data. The main contribution is the addition of weights to the previously developed ICDS approach, which allows for the correct modeling of segments in occlusion regions. The conducted experiments indicated that the proposed method is robust in modeling contours with occlusion, including curved segments and occlusions of different sizes. In terms of *F_score_* and *PoLiS*, the proposed method results in values of around 99% and 0.19 m, respectively. The drawback relates to the modeling of buildings with occlusions at or near the corners.

For future research, it is suggested to automatically estimate the weights for points in potential occlusion regions with the integration of data acquired from a different perspective, such as, for example, terrestrial LiDAR data or oblique images, in order to identify corners in occlusion regions. In addition, we suggest the use of an automatic approach for detecting the occlusion regions.

## Figures and Tables

**Figure 1 sensors-22-06440-f001:**
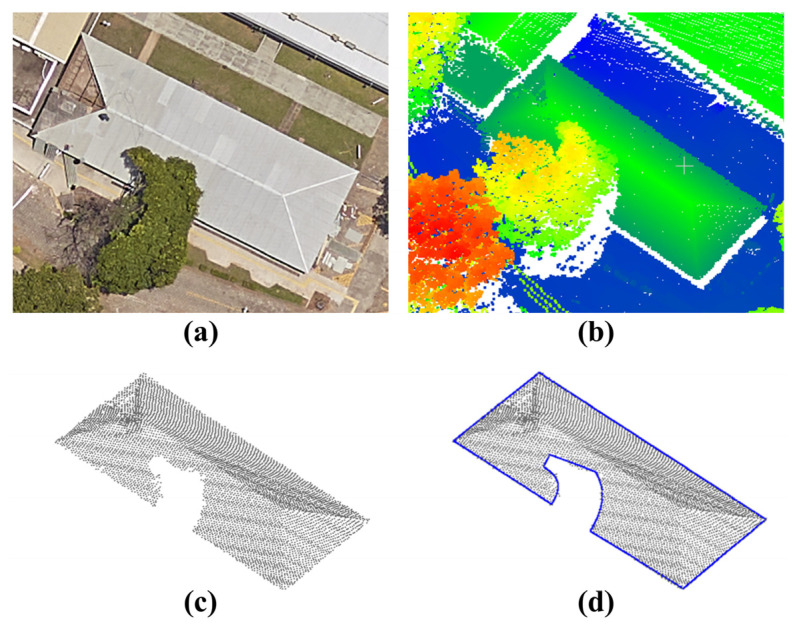
Building partially covered by a tree and modeled boundary using the ICDS method. Aerial image (**a**), airborne LiDAR data (**b**), and sampled points on the roof building (**c**) and modeled contour (**d**).

**Figure 2 sensors-22-06440-f002:**
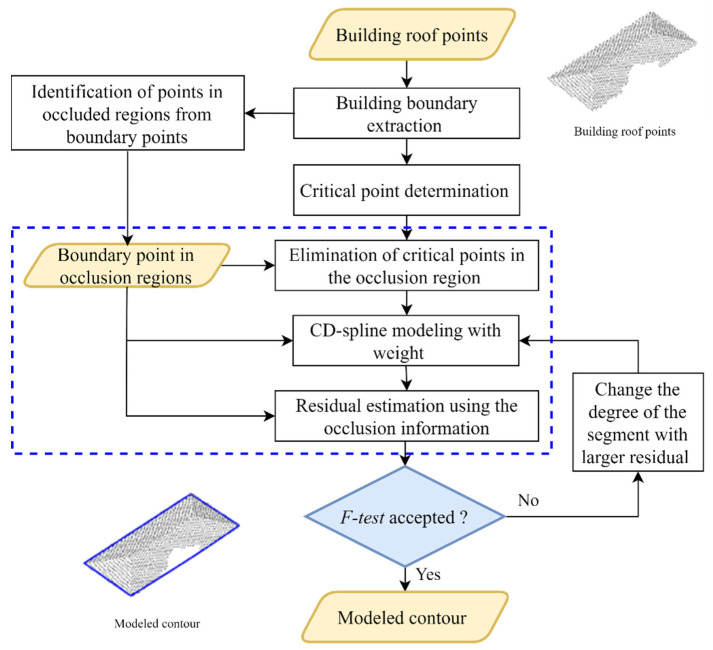
Flowchart of the proposed approach.

**Figure 3 sensors-22-06440-f003:**
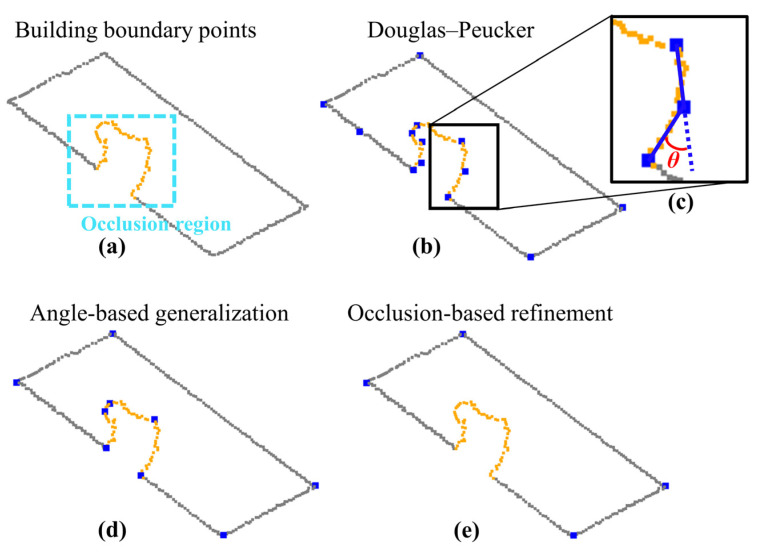
Critical point determination for a partially occluded building roof: Boundary points extracted using alpha-shape algorithm (**a**). Critical points derived from Douglas–Peucker (blue squares) (**b**). The angle *θ* between two adjacent lines formed by connecting adjacent critical points (**c**). Critical points derived from angle-based generalization (**d**), and from occlusion-based refinement (**e**).

**Figure 4 sensors-22-06440-f004:**
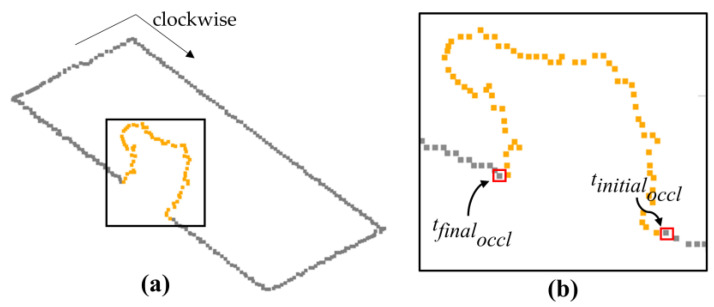
Building boundary with occlusion (**a**) and representation of the points related to $t_{{\mathrm{initial}}_{\mathrm{occl}}}$ and $t_{{\mathrm{final}}_{\mathrm{occl}}}$ (**b**). The orange points denote the contour points located in the occlusion region.

**Figure 5 sensors-22-06440-f005:**
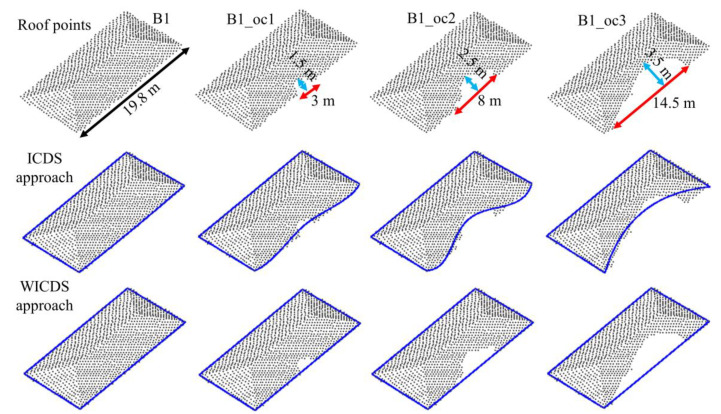
Rectangular building with different sizes of occlusion areas. Roof points (**First row**), modeled boundary (blue line) using ICDS (**Second row**) and WICDS (**Third row**).

**Figure 6 sensors-22-06440-f006:**
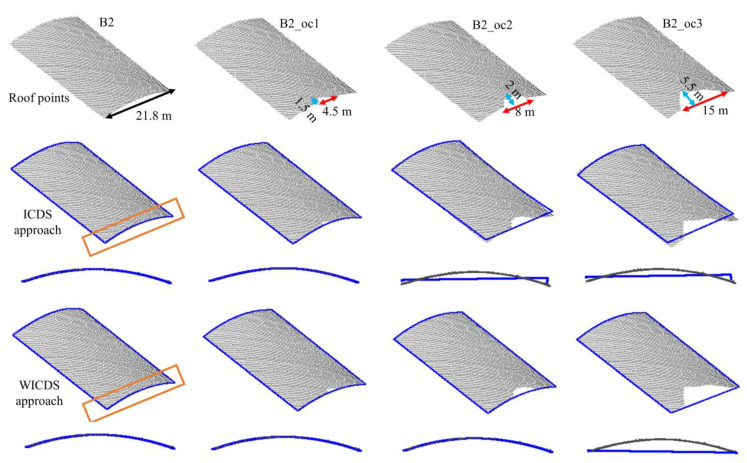
Curved building with different sizes of occlusion areas. Roof points (**First row**), modeled boundary (blue line) using ICDS (**Second row**) and WICDS (**Third row**).

**Figure 7 sensors-22-06440-f007:**
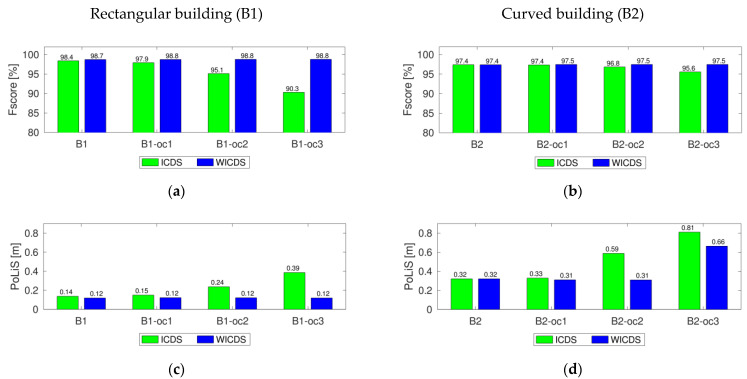
Quality metrics for rectangular (**a**,**c**) and curved buildings (**b**,**d**), considering the ICDS and WICDS methods.

**Figure 8 sensors-22-06440-f008:**
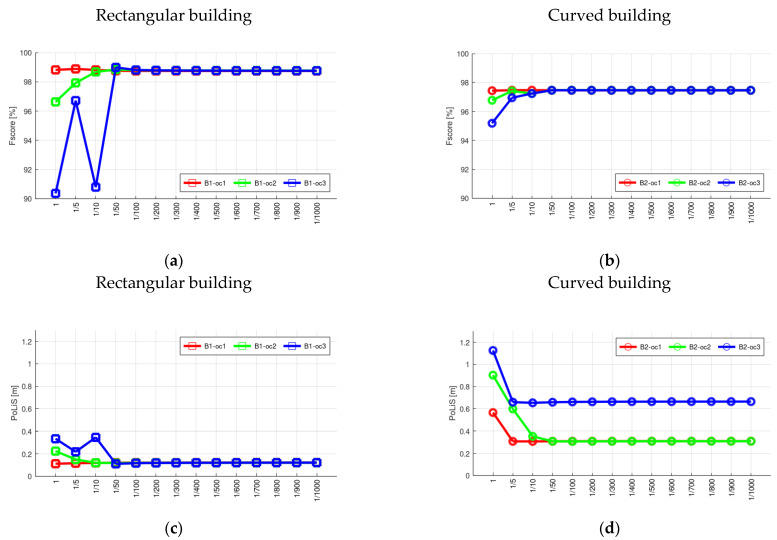
Quality metrics for rectangular (**a**,**c**) and curved (**b**,**d**) buildings with partial occlusions considering different weight values. *F_score_* (**a**,**b**) and *PoLiS* (**c**,**d**) metrics.

**Figure 9 sensors-22-06440-f009:**
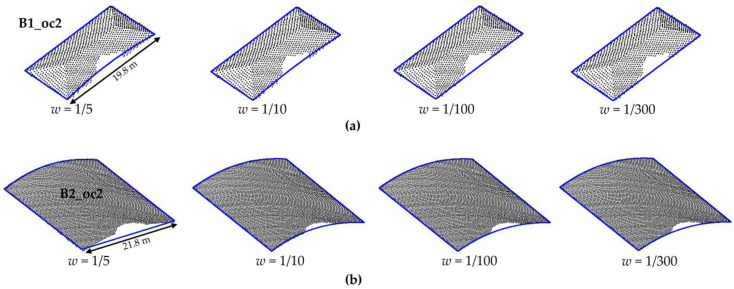
Modeled contour for buildings B1_oc2 (**a**) and B2_oc2 (**b**) using different weight values.

**Figure 10 sensors-22-06440-f010:**
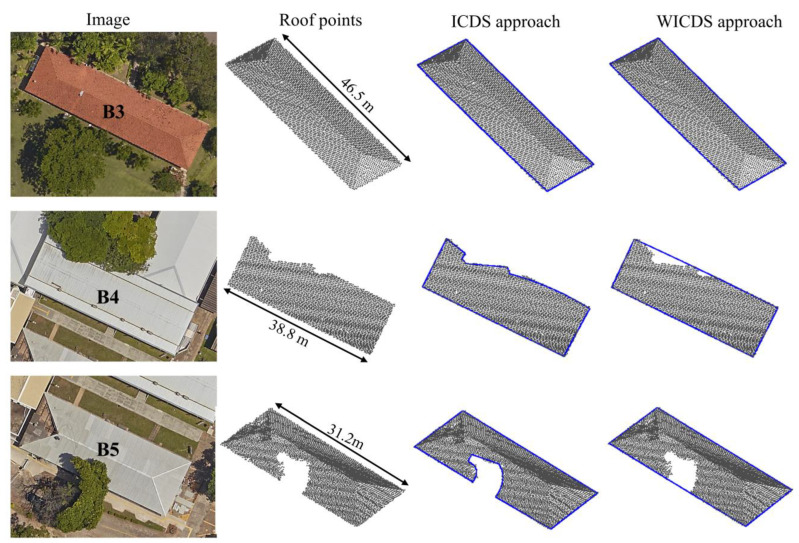
Buildings with occlusions selected in the Presidente Prudente/Brazil dataset. Aerial image patches (**first column**), points sampled over the building roof (**second column**), and results derived from ICDS (**third column**) and WICDS method (**fourth column**).

**Figure 11 sensors-22-06440-f011:**
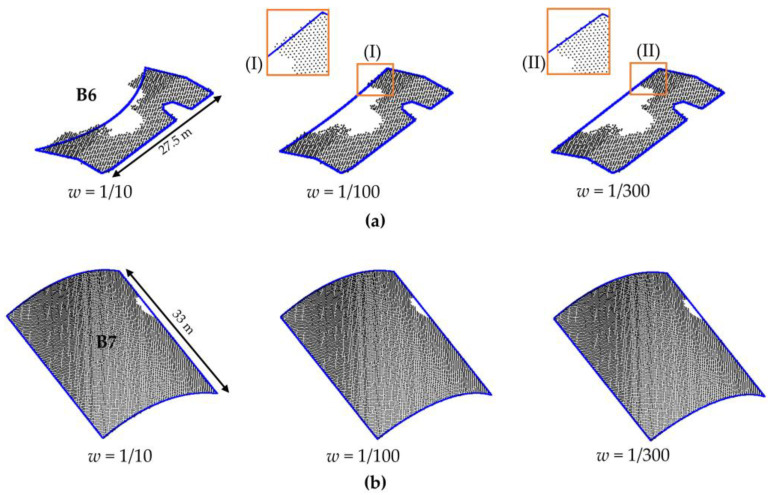
Modeled contour for buildings B6 (**a**) and B7 (**b**) using different weight values.

**Figure 12 sensors-22-06440-f012:**
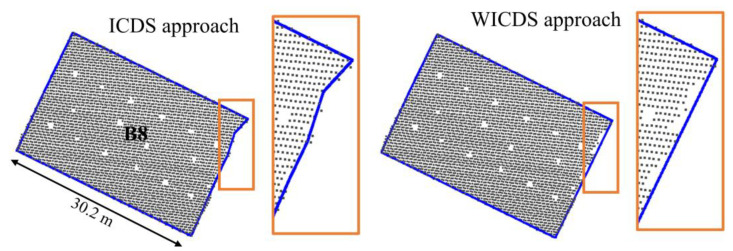
Modeled boundary for building B8. Results using the ICDS and WICDS method. The orange rectangles highlight the occlusion region.

**Figure 13 sensors-22-06440-f013:**
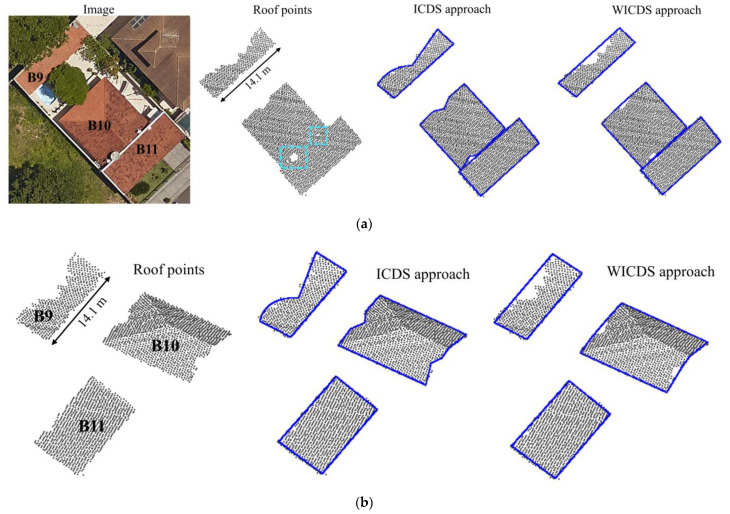
Two-dimensional (**a**) and three-dimensional (**b**) representations for buildings B9–B11. First row in (**a**): aerial image patches, roof points and results derived from building modeling methods. Second row in (**b**): representation 3D of roof points and results of boundary modeling. The cyan rectangles in (**a**) highlight the occlusions caused by antennas.

**Figure 14 sensors-22-06440-f014:**
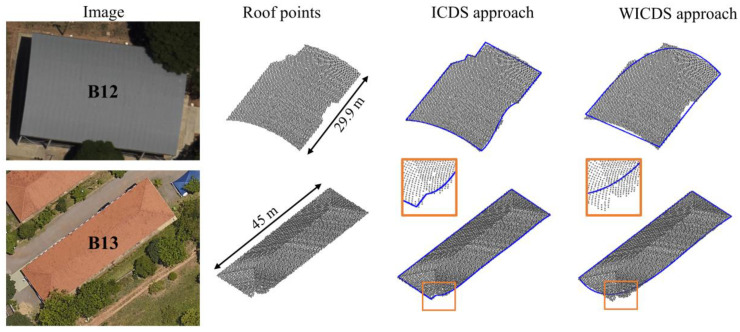
Occlusions at building corners caused by nearby trees. Building with curved segments (**first row**). Building with straight-line segments (**second row**). For both buildings, we show aerial image patches, roof points, and modeled boundaries. The orange rectangles highlight the corner region in B13 where the occlusion occurs.

**Figure 15 sensors-22-06440-f015:**
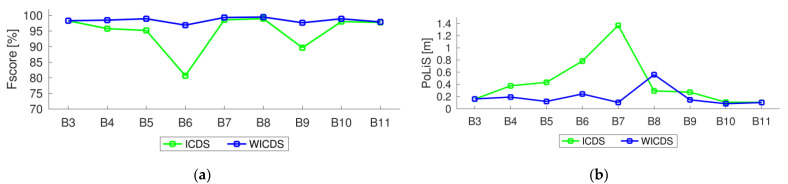
Quality metrics for buildings B3–B11 using the ICDS and WICDS methods. Plots of *F_score_* (**a**) and *PoLiS* (**b**) metrics.

**Table 1 sensors-22-06440-t001:** Estimated area and relative error in area using ICDS and WICDS approaches.

	Reference	ICDS Approach		WICDS Approach	
ID	Area (m^2^)	Area (m^2^)	*E_R_* (%)	Area (m^2^)	*E_R_* (%)
B3	595.72	584.24	−1.93	584.90	−1.82
B4	454.03	422.09	−7.03	444.76	−2.04
B5	383.05	349.42	−8.78	376.54	−1.70
B6	303.75	210.03	−30.85	290.91	−4.23
B7	805.95	801.49	−0.55	811.37	0.67
B8	612.09	601.90	−1.66	608.55	−0.58
B9	52.91	44.01	−16.82	51.76	−2.17
B10	108.84	106.80	−1.87	109.67	0.76
B11	77.00	75.32	−2.18	75.64	−1.77

**Table 2 sensors-22-06440-t002:** Quality metrics for different buildings using the ICDS and WICDS approaches.

	ICDS Approach				WICDS Approach			
ID	*Comp.* (%)	*Corr*. (%)	*F_score_* (%)	*PoLiS* (m)	*Comp.* (%)	*Corr*. (%)	*F_score_* (%)	*PoLiS* (m)
B3	97.4	99.3	98.3	0.159	97.4	99.2	98.3	0.162 (1.88%) *
B4	92.4	99.5	95.8	0.377	97.5	99.5	98.5	0.193 (−48.80%)
B5	91.0	99.8	95.2	0.434	98.1	99.8	99.0	0.121 (−72.12%)
B6	68.2	98.7	80.7	0.783	94.9	99.0	96.9	0.244 (−68.84%)
B7	98.3	98.9	98.6	1.368	99.7	99.0	99.4	0.105 (−92.32%)
B8	98.2	99.8	99.0	0.294	99.2	99.8	99.5	0.561 (90.82%)
B9	82.1	98.7	89.7	0.274	96.6	98.8	97.7	0.147 (−46.35%)
B10	97.1	99.0	98.1	0.109	99.3	98.6	98.9	0.084 (−22.94%)
B11	96.7	98.9	97.8	0.107	97.1	98.8	97.9	0.104 (−2.8%)
**Mean**	**91.3**	**99.2**	**94.8**	**0.434**	**97.8**	**99.2**	**98.5**	**0.191 (−55.93%)**

* Percentage of improvement (negative values ”−“) or deterioration (positive values “+”) of *PoLiS* metric for WICDS approach in comparison to the ICDS approach.

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
