# Peer review of "Weighted Iterative CD-Spline for Mitigating Occlusion Effects on Building Boundary Regularization Using Airborne LiDAR Data"

_sensors, 2022, doi:10.3390/s22176440_

Round 1
Reviewer 1 Report
This paper focus on the extraction of the regularization of the building boundary using the CD spline.
1, One of the most important issues in this paper is how to dealing with the occlusions in point cloud. However, the authors detect the possible occlusions manually.
2, One of the most important question in the building boundary regularization algorithm using CD-spline is how to get a serial of critical points as shown in Figure 2.
Once one or more critical points are not correct, they will affect the accuracy or the correctness of the subsequent boundary regularization.
Considering there may exist different complex cases in real datasets, can the method used in the paper (section 2.1) can always give correct critical points? A more detailed discussion on this question is necessary.
3. A weighted scheme is proposed to the existing ICDS, it can improve the accuracy of the building boundary.
4. The WICDS is one algorithm using CD spline, and the CD-spline algorithm is in 2D space. therefore the boundary given by WICDS is also in 2D space. However, the PoLiS metric used in this paper is in 3D space. Therefore, is it reasonable using a 3D metric to evaluate a 2D regularized building boundary?
Reviewer 2 Report
The manuscript entitled "Boundary regularization of buildings with partial occlusions from airborne LiDAR data using
a weighted iterative CD-spline" addresses an interesting and important topic for urban planning through data obtained by a promising tool, the LiDAR sensors. The article is well written, showing plan writing and good organization. However, despite using data obtained from LiDAR sensors, the manuscript focuses on developing a new method for processing this data, i.e., the focus is mathematical and statistical with applications to urban planning, not explicit in sensors.
I read the first references listed in this article, dos Santos, R.C.; Galo, M.; Habib, A.F. Regularization of Building Roof Boundaries from Airborne LiDAR Data 424 Using an Iterative CD-Spline. Remote Sensing 2020, 12, 1904, it seems that the authors sought to improve the previously proposed method for representing roof boundaries with occlusions by weighting the Iterative CD-Spline approach.
The cited paper, however, presents the methodological development in detail, showing the step-by-step process, which is not the case in the second manuscript. In addition, other issues draw my attention. 1. The title of the two articles is practically the same. 2. The first author is cited in several other references; 3. Despite significantly improving the prediction of roof boundaries in different cases, the quality metrics do not present such explicit results.
Since there is no problem determining the roof area with no occlusion, why not segment and analyze only the occlusion region? I believe the methodological appeal and quality metrics will bring more robustness to the results if assessing only the occlusion region. Furthermore, I assume the authors will have no problem creating an additional step within the programming logic to determine this segment and then evaluate the improvement separately.
Since this paper is an improvement of the iterative CD-Spline, I suggest that the authors create a flowchart similar to Figure 2 of the article dos Santos, R.C.; Galo, M.; Habib, A.F. Regularization of Building Roof Boundaries from Airborne LiDAR Data 424 Using an Iterative CD-Spline. Remote Sensing 2020, 12, 1904, intending to highlight the proposed new step. Furthermore, I believe the audience will highly benefit from a description of the steps followed during data processing (e.g., the programming script).
Furthermore, the article has merit to be published in a journal as a scientific article. However, some changes are necessary, mainly regarding its title, the repeated citation of articles previously published by the authors, and the detailed description of the previously published methodology changes.
Reviewer 3 Report
Dear authors, please find my observations in the attached document.
Thank you.

Round 2
Reviewer 1 Report
As to the question “boundary regularization of building with partial occlusions”: Considering the occlusion is detected manually, it is to say, to some extent, the occlusion area in the point cloud of the building is determined. Therefore, it is not significant to introduce a weight into ICDS method. The yellow points in Figure 4b is in occlusion region, it could not give any useful information for the building boundary obviously. Perhaps the easiest way is to delete them directly
As to the proposed WICDS algorithm itself, the new algorithm is indeed valuable and also can improve the accuracy of building boundary. However, it is not a solution to the question “boundary regularization of building with partial occlusions”. It may be more appropriate to remove "with partial occlusions" in the title.
